# OpenReview forum: "Diverging Preferences: When do Annotators Disagree and do Models Know?"
_ICML.cc/2025/Conference — ICML 2025 poster_

### Official Review · Reviewer_QJAM · 2025-03-13

**Overall Recommendation:** 4

**Summary:**

This paper investigates when and why human annotators disagree, identifying 4 broad sources of preference divergence (task underspecification, response style, refusals, errors) covering >30% of responses in RLHF datasets. They then explore how divergent preferences impact LLM training (with reward modelling) and evaluation (with LLM-as-Judge), showing that BT and LLM-as-Judge paradigms both fail to distinguish between instances of unanimous agreement and majority opinion. Arguing that divergent preferences are useful training signals (not undesirable noise), the authors develop a distributional reward modelling technique to utilise all preferences during training to model a distributional reward, and improve LLMs’ abilities to model human disagreements. Lastly, they point out deficiencies in the LLM-as-Judge evaluation pipeline, which punishes pluralistically aligned models by always expecting the modal responses (even when humans disagree), with consequences of penalising safety guardrails and clarification requests on underspecified prompts.

**Claims And Evidence:**

This paper presents 3 central claims: 1) human annotators meaningfully disagree in RLHF datasets such that these divergences are useful training signals (not noise), 2) BT reward modelling fails to differentiate between unanimous versus majority rule and falls short of pluralistic alignment, 3) LLM-as-Judge evaluation techniques suffer similar pitfalls and penalise pluralistically aligned LLMs with safety guardrails and which know to ask for clarification on underspecified prompts. All claims are well evidenced with experiments, concrete data examples, thorough analysis, both numerical and qualitative results.

**Essential References Not Discussed:**

The related work section is thorough and satisfactorily engages with relevant work.

**Experimental Designs Or Analyses:**

Training stage comparisons between the proposed Mean-Var Reward Models (KL), scalar-valued reward modelling (BT, MSE-Regression, Skywork, 70B-Reward) and Mean-Var Baseline (NLL, Independent) are thorough, well-documented and well-structured. Analysis is granular and discusses in detail the different sources, types and degrees of disagreement; this closely matches the proposed method, which does not assume independence of judgement and models the mean and variance of the preference distribution to capture the shape/spread (strength and spectrum) of opinions. Discussion on the limitations of LLM-as-Judge as similarly nuanced and reveal future directions for improvement.

**Methods And Evaluation Criteria:**

The training proposal is evaluated on extended versions of MultiPrefDisagreements and HelpSteer2-Disagreements datasets, for a variety of frontier LLMs. Evaluations are thorough with results reported for both preference accuracy and diverging ID AUROC metrics, which respectively capture alignment to the modal preference and pluralistic alignment. LLM-as-Judge evaluation is benchmarked on ChatbotArena (Arena-Hard) and is shown to unfairly penalise pluralistically aligned models when given a divisive example. The method and evaluation procedures are sound and reasonable.

**Other Comments Or Suggestions:**

The following are more suited as comments or out-of-interest questions; discussing them could lead to a score increase but not addressing them fully will not lead to a score decrease:
- The authors might consider engaging further with classic literature on social choice and voting theory, e.g. Arrow's impossibility theorem and the inability to simultaneously satisfy intuitive desiderata when attempting to model a spectrum of preferences with a social utility function.
- Do the datasets also provide demographic attributes of the annotators? If so, are disagreements (variance in opinions) more pronounced inter (between) or intra (within) demographic groups?
- How does distributional reward modelling compare to 1) training only on the 70% of non-diverging opinions, and to 2) in-context learning or context-aware techniques?

**Other Strengths And Weaknesses:**

To summarise the above, this paper is a worthwhile contribution that will hopefully pave the way for further work in pluralistic alignment. It is significant (with dataset and method contributions), clearly presented and soundly argued. While the idea that annotators disagree, or that reward modelling does not capture the full spectrum of human perspectives has been explored before, the existence of prior intuition does not detract from the merits of this work since it ventures beyond mere contemplation to contribute an original distributional reward modelling approach for LLMs. One possible weakness of this work is the lack of formal investigation of how BT or LLM-as-Judge collapses to the modal opinion, though this is likely beyond the scope of this article.

**Questions For Authors:**

N/A

**Relation To Broader Scientific Literature:**

1. **Important dataset releases (of individual preferences) -** The authors collaborated with creators of MultiPrefDisagreements and HelpSteer2-Disagreements to release the individual annotator preferences (before aggregation) to enable further work on training and evaluation under human disagreements.
2. **Significant and relevant -** This submission relates to frontier directions on modelling the preferences of different users and adapting LLM responses to be more helpful, user-aligned and target-specific.
3. **Important implications for benchmarking -** This paper confirms that mainstream LLM-as-Judge evaluation pipelines are mismatched with many desiderata (e.g. pluralistic alignment, AI safety, clarity of thought) of LLMs, and will hopefully motivate further research and advances in robust evaluation.

**Theoretical Claims:**

N/A: This paper does not make theoretical claims.

---

> ### Author Rebuttal · Authors · 2025-03-31
>
> Thank you for your thoughtful review and suggestions!
>
> **Q1: The authors might consider engaging further with classic literature on social choice and voting theory.**
>
> Thank you for this suggestion! We agree that Arrow's impossibility theorem and social choice theory is relevant to our work, particularly to the discussions in Section 5 on pluralistic alignment and different practitioners' decisions in comply/refuse disagreements. In our revisions, we will add include such works in these discussions
>
> **Q2: Do the datasets also provide demographic attributes of the annotators? If so, are disagreements (variance in opinions) more pronounced inter (between) or intra (within) demographic groups?**
> We include annotator details in Appendix C. To summarize from there, these datasets do not include demographic information or annotator IDs due to privacy concerns. We are therefore unable to conduct such an analysis.
>
> **Q3: How does distributional reward modelling compare to 1) training only on the 70% of non-diverging opinions, and to 2) in-context learning or context-aware techniques?**
> We do not perform the first experiment; however, we do not expect it to have significant differences in Bradley-Terry or MSE-Regression reward models when trained with all or aggregated annotations. Training on all annotations, in particular, is perhaps closest to this setting, as the diverging and non-diverging examples are differentiated during training.
> Regarding your second question, our goals with our LM-as-Judge experiments were intended to assess the behavior of existing, standard LM-as-Judge benchmarks. We do not experiment with different prompting techniques to improve the LM-Judge’s predictions, but agree that this is an exciting area for future work and will add discussion for it along with our other results and recommendations (Section 5.3) in our revisions.
>
> ---
>
> Thank you again for your suggestions, and please let us know if you have any remaining suggestions or questions.

---

> > ### Comment · Reviewer_QJAM · 2025-04-08
> >
> > Thank you for the detailed response. I maintain that this work makes valuable contributions towards understanding (limitations of present) reward modelling for LLMs. It in fact goes one further to propose a dataset and method to better enable pluralistic alignment. I believe my original score of 4 (accept) is befitting of the manuscript's quality and potential (for facilitating future investigations); I maintain my score and affirm my belief that it is worthy of acceptance.

---

### Official Review · Reviewer_BbMo · 2025-03-14

**Overall Recommendation:** 2

**Summary:**

- This paper introduces two datasets consisting of annotations for potential reasons for disagreements in preferences and derives a taxonomy from these annotations.
- The paper also studies the distribution of rewards across different modeling techniques.
- The paper also compares single versus distributional reward modeling methods.
- The paper finally evaluates LLM-as-a-judge in divisive examples.

## update after rebuttal
The authors have addressed most of my comments. I remain concerned that the experiments across different portions of the paper were conducted on varying datasets, limiting the potential generalizability of the work.

**Claims And Evidence:**

Overall, the organization of the paper and thus the main point was challenging to follow. It was almost as if Sections 3-5 were each their own papers glued together. There was no contributions list in the work to evaluate whether claims were appropriately substantiated.

**Essential References Not Discussed:**

N/A

**Experimental Designs Or Analyses:**

More detailed questions about experimental design:
- Section 2: How do you separate “style” (in taxonomy) from noise? Do annotators have the right background to understand and correctly determine the cause for diverging preferences?
- Section 3: Can you show Figure 2 for the other models, as this is only Bradley Terry? To what extent are these trends dataset-specific? The text doesn’t discuss that High-Agreement Ties do not follow the same trend for Figure 2.
- Section 4: Can you provide error bars for Table 3? Preference accuracy is not that different between single-value vs distributional reward models.
- Section 5: The AUROC of the KL model trained was still relatively low, which means that it is a very imperfect classifier to use on WildBench data. Were any checks done to verify that the model is useful? The authors report results over 50 examples, do these results generalize more broadly?

**Methods And Evaluation Criteria:**

- The paper needs to make clear where the individual datasets (e.g., MultiPref and HelpSteer2) come from and appropriately cite the authors when the datasets themselves are first mentioned. Relatedly, I think it’s also very disingenuous to say that this paper introduces two datasets when the only thing that it did was to subset for disagreement.
- Evaluations across sections draw from a lot of different datasets and models, making it hard to follow where potential confounding issues might stem from (e.g., data or model-specific choices).

**Other Comments Or Suggestions:**

N/A

**Other Strengths And Weaknesses:**

N/A

**Questions For Authors:**

Please address aforementioned experimental design questions.

**Relation To Broader Scientific Literature:**

There is increasing work on pluralistic alignment. This work aims to shed light on potential issues that may arise when performing reward modeling or using LLM-as-a-judge.

**Theoretical Claims:**

N/A

---

> ### Author Rebuttal · Authors · 2025-03-31
>
> Thank you for your feedback! We address each comment below. Please let us know if you have any remaining questions or concerns.
>
> **Clarifying Dataset Collection and Release**
> HelpSteer2 and MultiPref are cited. Note that MultiPref was made public prior to the publication documenting it, thus we communicated with the dataset authors to determine how to cite it for this submission. We will update these citations in our revisions.
> We also make it explicit that we do not collect annotations ourselves for this work on Page 1: “Note that we did not collect new datasets but instead are releasing the individual annotations of these existing datasets (which previously released only annotations aggregated across multiple annotators for the same task), with support from the dataset creators.”
>
> **Contributions List & Connections between Contributions**
> See our response to Reviewer 1 under **“Contribution List”**.
>
> **Sec 2: How do you separate “style” (in taxonomy) from noise?**
> When categorizing disagreement causes, we label instances with specific subclasses (i.e., Verbosity, Format, Complexity, Aesthetic Tastes). The “style” meta-category is used to describe subclasses where the responses do not differ in their interpretation of the prompt nor their high-level content, rather how the information is presented to the user. Below, we paraphrase our definitions of each “style” subclass:
> * **Style:** Where instances are labeled if both responses interpret the prompt similarly, but…
>   * **Verbosity:** … differ in their level of detail or in including supplementary examples.
>   * **Format:** … differ how they organize their responses under lists or headings.
>   * **Complexity:** … are targeted toward users with different levels of domain-expertise (e.g., technical jargon that appears in only one response).
>   * **Aesthetic Taste:** … the prompt is open-ended for generation and differences are primarily in style / tone / creative choices.
>
> **Sec 3: Figure 2 for other models/datasets**
> These visualizations for all 8 models+datasets in Table 2 are in the Appendix (Figures 4 & 5). As Table 2 summarizes, we find similar trends across all settings.
>
> **Sec 3: Figure 2 – why do High-Agreement Ties not follow the same trend?**
> We expect opposite behaviors from reward models on High-Agreement Preferences and Ties. For High-Agreement Preferences, we expect reward models to predict a large gap in rewards of the two responses. For High-Agreement Ties, we expect reward models to predict a small or no gap in the rewards.
>
> **Our Table 2 results and visualizations (Figures 2, 4, & 5) demonstrate:**
> Standard reward models correctly capture this expected behavior on High-Agreement Preferences and Ties.
> Their predictions on examples with diverging preferences are indistinguishable from their predictions on examples with High-Agreement Preferences. They predict clear preference for one response in cases of annotator disagreement (further supported in later from Table 3 results).
>
> **Sec 4: Preference accuracy is not that different between single-value vs distributional reward models (Table 3).**
> We do not claim a large difference in preference accuracy. In our results discussion, we state “We find that, with the exception of the Mean-Var (NLL, Indep.) baseline, all systems perform comparably in Preference Accuracy.” (Section 4.2)
> Our goal with proposing distributional reward models is not to improve preference accuracy, but rather to develop a reward model that can identify diverging preferences without compromising on preference accuracy.
>
> **Sec 5: Utility of the relatively low AUROC mod on WildBench data & Generalization**
> Our systems are imperfect at identifying diverging preferences. As such, we suggest that they can be used to assist benchmark authors “by identifying divisive prompts in LLM-as-Judge benchmarks so they can be further examined by benchmark authors and removed.” (Section 5.3) While future work may develop better methods for identifying divisive examples, our experiments are a proof of concept for such an approach.
>
> We examine the 50 most divisive examples identified by our model in Wildbench (an out-of-distribution dataset) and find that the majority of these instances are divisive prompts where the task is ambiguous or where it may be reasonable for systems to refuse or comply with the request, depending on the LLM developer’s specifications. We, furthermore, find that the LLM-Judge scores responses that interpret the request differently or decide to comply/refuse substantially differently on these examples. We provide examples of such instances from Wildbench in the Appendix (Tables 7 and 8).
>
> To further support this, we repeat this analysis over 700 sampled instances from ChatbotArena and analyzed the top 30 most divisive examples as identified by our system. And similarly found that 17 were similarly divisive (ambiguous or reasonable to comply/refuse). We will include this analysis and provide examples in our revisions.

---

### Official Review · Reviewer_Vr8U · 2025-03-14

**Overall Recommendation:** 3

**Summary:**

The authors examine diverging preferences in human-labeled datasets and present a taxonomy of disagreement sources. They show most disagreements stem from task underspecification and response style, not annotator errors, challenging the view that disagreements are mere noise. Standard reward modeling methods, like the Bradley-Terry model, overlook the difference between unanimous agreement and majority opinions, undermining pluralistic alignment. To address this, they propose methods to identify and mitigate diverging preferences in evaluations and training, promoting better LLM alignment.

**Claims And Evidence:**

Yes.

**Essential References Not Discussed:**

References are properly discussed.

**Experimental Designs Or Analyses:**

Experimental design is reasonable.

**Methods And Evaluation Criteria:**

Yes.

**Other Comments Or Suggestions:**

Refer to *Other Strengths And Weaknesses* section.

**Other Strengths And Weaknesses:**

**Strengths**
- The paper provides a valuable analysis of diverging preferences in language model alignment by developing a clear taxonomy of disagreement sources and identifying task underspecification and response style as primary causes.
- The proposed distributional reward model effectively captures diverging preferences, with experimental support.
- The identification of bias in LLM-as-Judge evaluations on leading benchmarks offers meaningful insights for the community, enhancing the understanding of benchmark results.

**Questions For Authors:**

- Can you clarify the rationale behind selecting the specific values used to map the reward gap to various annotator preferences? (l247-255, right column)

**Relation To Broader Scientific Literature:**

This paper is related to the disagreement study in NLP domain and model-based reward modeling.

**Theoretical Claims:**

No theoretical results.

---

> ### Author Rebuttal · Authors · 2025-03-31
>
> Thank you for your thoughtful review and positive feedback.
>
> **Q: Can you clarify the rationale behind selecting the specific values used to map the reward gap to various annotator preferences?**
>
> Appendix A provides more details on how we select these hyperparameters (we select the best performing value on development data). Rationale for selecting values is twofold: (1) MSE regression systems support that mapping rewards / preferences to linearly spaced intervals has strong performance and (2) Unlike “slight preference” or “tied” judgements, “significant preference” judgments may represent an arbitrarily large gap between the quality of the the two responses, hence we allow the gap between rewards for such responses to also be unbounded.
>
> ---
>
> Thank you again for your careful review of our paper and feedback! Please let us know if we have addressed your remaining concerns or questions. We also would appreciate any additional suggestions or clarifications that might improve your assessment.

---

### Official Review · Reviewer_emHx · 2025-03-17

**Overall Recommendation:** 3

**Summary:**

This paper investigates diverging preferences in human-labeled datasets used for reward modeling and language model evaluations. The authors develop a taxonomy of disagreement sources. Through empirical analysis of HelpSteer2 and MultiPref, they find that disagreements are not random noise but stem from systematic differences in annotator preferences. The paper further demonstrates that standard reward models fail to distinguish between high-agreement and diverging preferences. To address this, the authors propose a Mean-Variance reward model with KL-divergence training, which captures the distribution of annotator preferences. Additionally, the paper studies the impact of diverging preferences of popular LLM-as-Judge methods for evaluating LLMs and proposes a method for removing instances of diverging preferences in LLM-as-Judge benchmarks.

**Claims And Evidence:**

The first main claim is that diverging preferences are prevalent and not annotation noise. **Evidence:** The authors analyze preference pairs from two datasets, showing that 30-39% of them contain diverging preferences. They further categorize the reasons behind disagreement and provide statistics for each category.

The other claim is existing reward models treat diverging preferences similarly to high-agreement preferences. The proposed Mean-Variance Reward Model better captures human preference distributions. **Evidence:** Experiments on Bradley-Terry and MSE Regression models show that they predict nearly identical reward differences for high-agreement and diverging cases. This holds even when trained on all annotator labels instead of aggregated preferences. Experiments show that the proposed Mean-Variance model improves Diverging ID AUROC by 0.16 over standard reward models, indicating better identification of diverging preferences.

**Essential References Not Discussed:**

Not identified.

**Experimental Designs Or Analyses:**

The experimental design makes sense and the results seem to be solid.

**Methods And Evaluation Criteria:**

The proposed Mean-Variance model and evaluation criteria make sense to me.

**Other Comments Or Suggestions:**

See the Questions below.

**Other Strengths And Weaknesses:**

One weakness is the taxonomy of disagreement sources developed in the first half of the paper seems to be distached from the second part where the new algorithm is proposed. The mean-variance model does not benefit from the taxonomy defined, only the empirical distribution of disagreement in data makes a diffference.

**Questions For Authors:**

In sectioni 4, training Mean-Var reward models requires a map between the reward difference and the preference annotations such as slightly preferred, significantly preferred, based on what the range of reward difference such as  (−0.5, 0.5),  [0.5, 1.5) is picked? How do you chose the turning points, would changes of those points affect the model performance?

**Relation To Broader Scientific Literature:**

Focusisng on the disagreement in preferenece, especially differentiating clear preference and preference with ambiguity is novel compared to the prior literature.

**Theoretical Claims:**

No theorrm provided in the main text.

---

> ### Author Rebuttal · Authors · 2025-03-31
>
> Thank you for your thoughtful review and positive feedback!
> We would like to address and clarify the following points in the review:
>
> **W1: Contribution List + Connection between taxonomy and reward modeling**
> We agree that our contributions and their connections would be more clear. In our revisions, we will include a list of contributions and will strengthen the connections between the taxonomy and the experimental work to address this concern. To summarize, our goals and contributions are as follows:
> * **Goal 1:** Identify where disagreements in preference annotation come from.*
>   * **Contribution 1:** We analyze diverging preferences in two datasets and develop a taxonomy of causes. Contrary to standard modeling practices, we find the majority of disagreements are not influenced by the correctness or appropriateness. They are, instead, due to factors such as underspecified prompts, verbosity, etc. We further work together with the dataset creators to release individual annotator judgments to support future efforts studying diverging preferences.
> * **Goal 2:** Understand how the LLM development pipeline is affected by diverging user preferences, focusing on the two most directly impacted areas: reward modeling and evaluation.
>   * **Contribution 2:** We find that standard reward modeling approaches (e.g., Bradley-terry models) and evaluation methods (LLM-as-Judge) both fail to capture diverging user preferences by predicting a clear preference for a single response, even when annotators disagree.
> * **Goal 3:** Suggest novel methods for identifying examples with diverging preferences in reward models and evaluations.
>   * **Contribution 3:** We develop a novel distributional reward modeling method that achieves strong reward modeling performance while also outperforming existing methods at identifying examples with diverging preferences. We also demonstrate that this model can be used to identify problematic examples in LLM-as-Judge benchmarks where LM-Judges demonstrate strong preference toward a single type of response, even when annotators would disagree.
> Connection between taxonomy and reward modeling: The taxonomy (Contribution 1) serves as a necessary foundation to validate that the disagreements in our dataset are meaningful (and not just noise) before proceeding to model them (Contributions 2+3). Without establishing the types and patterns of disagreement first, any modeling efforts would lack proper grounding.
>
> **W2: Specific disagreement types Experiments**
> Regarding experimentation on specific disagreement types, we faced practical limitations. It’s hard to do experiments on specific subsets of disagreement types when we don’t have the ability to label many instances.
>
> **Q1: Based on what the range of reward difference such as (−0.5, 0.5), [0.5, 1.5) is picked? How do you choose the turning points?**
> Appendix A provides more details on how we select these hyperparameters (we select the best performing value on development data). These values are also based on observing that MSE regression reward models, which mapping rewards / preferences to linearly spaced intervals, have strong performance
>
> ---
>
> Thank you again for your careful review of our paper and your valuable feedback! Have we successfully addressed your concerns? We would appreciate any additional suggestions for clarification or modifications that might improve your assessment.

---

### Decision · Program_Chairs · 2025-05-01

**Decision:**

Accept (poster)

**Comment:**

This paper examines annotator disagreement in preference datasets used in large language model (LLM) development. It introduces a taxonomy of disagreement sources, evaluates how existing reward models and evaluation pipelines (such as LLM-as-Judge) fail to account for disagreement, and proposes a distributional reward modeling method that captures diverging preferences. The authors also release individual-level annotations for existing datasets and provide tools to identify divisive examples during training and evaluation.

Overall, the feedback from reviewers was positive. On the one hand, reviewers appreciated the paper’s focus on disagreement, the release of individual annotations, and the modeling approach tailored to pluralistic alignment. On the the other hand, reviewers pointed out issues in clarity and cohension. In particular, BbMo highlighted that the paper reads like several standalone sections, and questioned the value of the analysis. the proposed models achieve modest performance gains and rely on diverse datasets that make it difficult to track confounding variables.

Having read the reviews, the rebuttal, and the paper, I am recommending accept at this time – though I note that the decision is borderline. In this case, my recommendation is based on the fact that the paper makes a meaningful contribution by nature of the topic that it studies. The work highlighting the role of pluralistic disagreement and providing tools for identifying divisive examples. Looking forward, I believe that a potential camera-ready version should address two key issues (raised by BbMo)to ensure its contributions will be valuable to a growing stream work in pluralistic alignment. These includeL

1. Cohesion. The taxonomy, modeling, and evaluation sections feel disconnected. The final version should include a concise contributions list and a stronger narrative linking these parts into a unified story.

2. Confounding factors in experiments. Evaluations draw from many datasets and models. Add a discussion of dataset/model-specific effects and make clear which results generalize across settings.